# CA-STD: Scene Text Detection in Arbitrary Shape Based on Conditional Attention

Xing Wu [1,2,*,†] , Yangyang Qi [1,†], Jun Song [3] , Junfeng Yao [4], Yanzhong Wang [5], Yang Liu [1], Yuexing Han [1] and Quan Qian [1,2]

1   School of Computer Engineering & Science, Shanghai University, Shanghai 200444, China
2   Zhejiang Laboratory, Hangzhou 311100, China
3   Department of Geography, Faculty of Social Sciences, Hong Kong Baptist University, Hong Kong 999077, China
4   Cssc Seago System Technology Co., Ltd., Shanghai 200010, China
5   Shanghai Jianke Engineering Project Management Co., Ltd., Shanghai 200032, China
*   Correspondence: xingwu@shu.edu.cn
†   These authors contributed equally to this work.

**Abstract:** Scene Text Detection (STD) is critical for obtaining textual information from natural scenes, serving for automated driving and security surveillance. However, existing text detection methods fall short when dealing with the variation in text curvatures, orientations, and aspect ratios in complex backgrounds. To meet the challenge, we propose a method called CA-STD to detect arbitrarily shaped text against a complicated background. Firstly, a Feature Refinement Module (FRM) is proposed to enhance feature representation. Additionally, the conditional attention mechanism is proposed not only to decouple the spatial and textual information from scene text images, but also to model the relationship among different feature vectors. Finally, the Contour Information Aggregation (CIA) is presented to enrich the feature representation of text contours by considering circular topology and semantic information simultaneously to obtain the detection curves with arbitrary shapes. The proposed CA-STD method is evaluated on different datasets with extensive experiments. On the one hand, the CA-STD outperforms state-of-the-art methods and achieves 82.9 in precision on the dataset of TotalText. On the other hand, the method has better performance than state-of-the-art methods and achieves the F1 score of 83.8 on the dataset of CTW-1500. The quantitative and qualitative analysis proves that the CA-STD can detect variably shaped scene text effectively.

**Keywords:** scene text detection; conditional attention; contour information aggregation



## 1. Introduction

Scene Text Detection(STD) is required for many popular technologies including text recognition, picture understanding, and automated driving. The STD models are used to detect the location of text in a given image. Nowadays, the most advanced models rely on complex components and are only adaptive to rectangular text. However, uniform detection boxes contain much background information for the curved text, resulting in a lot of noise for text recognition.

In recent years, most methods have treated STD as object detection. Convolutional Neural Networks (CNN) are regarded as the feature extractor in these methods. Although STD models based on CNN performed well on a variety of datasets, they fell short in some critical, demanding cases, for instance, in-plane rotation, multi-directional and multi-resolution text, complex typeface, perspective distortion, occlusion, shadow, image blur, and a complex background. The disadvantage of these methods can be traced back to the inherent limitations of CNN, whose convolutional unit samples the characteristic map at a set place.

Therefore, we propose a method different from the existing ones for STD, which is based on a Transformer with conditional attention, as shown in Figure 1. Raisi et al. [1] introduced a traditional Transformer into their method to retrieve relevant information from an entire sequence by looking through vectors backward and forward. However, the method was not suitable for the curved text common in real life. To meet the challenge, the proposed CA-STD makes the following contributions:

- Designing the Feature Refinement Module (FRM) to enhance features with a lightweight network;
- Proposing the conditional attention mechanism to improve the model's performance;
- Generating the bounding boxes with arbitrary shape by the Contour Information Aggregation (CIA).

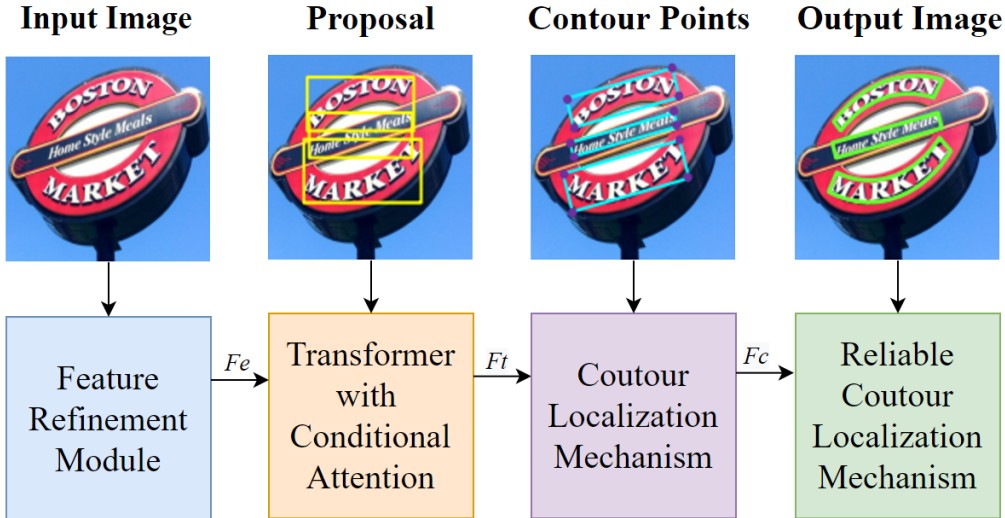

**Figure 1.** The overall pipeline of the proposed CA-STD.

## 2. Related Work

The related work and current state of research on Scene Text Detection (STD) will be presented in three aspects. Firstly, in terms of the datasets on which the models rely, STD models can be divided into two main categories: models based on character-level and word-level labels. Secondly, the performance of Convolutional Neural Networks (CNN) in STD cannot be underestimated. The representative methods are going to be introduced according to their optimization with different components. Last but not least, Transformer is currently in full swing and making a breakthrough in the image field. Therefore, the last part of this chapter will detail the application of the Transformer in STD.

According to the dataset labels, STD models can be categorized into two types. One is based on the character-level dataset, while the other is based on the word-level dataset. Zhang et al. [2] suggested a STD model based on the Maximum Stable Extreme Regions (MSER) [3]. However, the model did not work for some images with low contrast between the background and the text [4,5]. After that, Yao et al. [6] proposed a model, incorporating a character map and word map to jointly detect text, whose performance was determined by the character-level labels. Shi et al. [7] presented SegLink, a method cutting words into small text blocks easier to detect and then predicting neighboring connections to join the small text blocks into words. It is notable that the excellent performance of the segmentation network is the prerequisite to ensuring the model's performance. P. Lyu et al. [8] proposed Mask Text Spotter, which fulfilled STD by predicting the probability map of characters. However, the algorithm required datasets at a high level. Deng et al. [9] suggested to eschew border regression in favor of segmenting the scene text as an instance and then finding the rectangular box of the matching text directly. Wang et al. [10] proposed PSENet, which detected related targets by kernel clustering. Long et al. [11] proposed to detect

scene text by predicting the text area and centerline and adding geometric attributes to the algorithm. Ye et al. [12] proposed TextFuseNet, which is based on character-level labels.

After the CNN were used to extract features, the performance of the STD model began to depend on the design of special components, like Region Proposal Network (RPN), Feature Pyramid Network (FPN) [13,14], anchors, and other factors [15,16]. These algorithms required a lot of prior knowledge and complex post-processing steps. They generated many candidate boxes according to anchors, made a series of adjustments, and finally obtained the prediction boxes through Non-Maximum Suppression (NMS). Therefore, Single Shot MultiBox Detector (SSD) [17,18] and Recurrent Neural Network (RNN) [19] improved the model's performance by designing the anchors. Although Liao et al. [20] made extensive use of rotation invariance capabilities to detect scene text, it was still not capable of detecting text with arbitrary shape. TextBoxes [21] detected text by modifying the convolutional kernel and anchor boxes, which is adaptive to various scene text. Rotation Region Proposal Network (RRPN) [22], inherited from Fast R-CNN [23], could also detect scene text with arbitrary shape. Wang et al. [24] proposed ContourNet, whose performance depended on the design of candidate boxes. Du et al. [25] proposed to extract the feature of gap in images to detect arbitrarily shaped text, which relies on a large number of conventional CNN components.

Nowadays, more and more research about images introduces the Transformer and abandons traditional CNN [16]. Vision Transformer (ViT) [26] improved Transformer to classify images. Detection Transformer (DETR) [27] adopted Transformer to detect and recognize objects. In Generative Pretraining from Pixels (Image GPT) [28], images were completed by a Transformer. Chen et al. [29] offered a Transformer-based detection model for automatic driving that seeks to recognize lane marking. The initial application of Transformer in scene text detection, Transformer-based Text Detection in the wild [1], made a breakthrough but failed to detect curved text common in natural scenes.

## 3. Method

CA-STD is proposed to detect scene text with arbitrary shapes. In the method, a Feature Refinement Module (FRM) is designed to increase the receptive field of the feature map. After that, the proposed method is based on the Transformer rather than Convolutional Neural Networks (CNN) common in traditional detection method. Furthermore, traditional Transformer is based on an attention mechanism, which pays attention to the global features and then maps almost consistent attention weights to the feature map. On the one hand, it slows the model's convergence. On the other hand, the influence of location information is ignored. The improved Transformer provides a conditional attention module that decouples spatial and content information provided by the image to generate horizontal detection boxes. Finally, the horizontal detection boxes are regressed step by step to obtain an arbitrarily shaped curve matching the scene text.

### 3.1. Feature Extraction Network

ResNet50 is chosen as the backbone in CA-STD because of its parameter effectiveness and ability to alleviate gradient disappearance. To reduce the amount of calculation, the convolutional kernel $1 \times 1$ is used to downscale the dimensions of the feature map. The output map is refined by FRM as shown in Figure 2, where $s$ denotes the stride of convolution. FRM is organized in the shape of 'U', which includes two parts (upper and lower sampling part). The upper sampling part takes the multi-layer feature map from CNN as the input, and the latter takes the output from the last step as its input. The enhanced feature map is the same as the original in the size.

$$F = FRM(B(X)) \tag{1}$$

where $X \in \mathbb{R}^{W \times H \times 3}$, and $X$ is the input image. $B$ represents the backbone, which is ResNet50 in our proposed method. FRM is a feature extraction module. $F \in \mathbb{R}^{W_i \times H_i \times d}$, where $i$ and $d$ index the number of feature layers and channels, respectively, is enhanced.

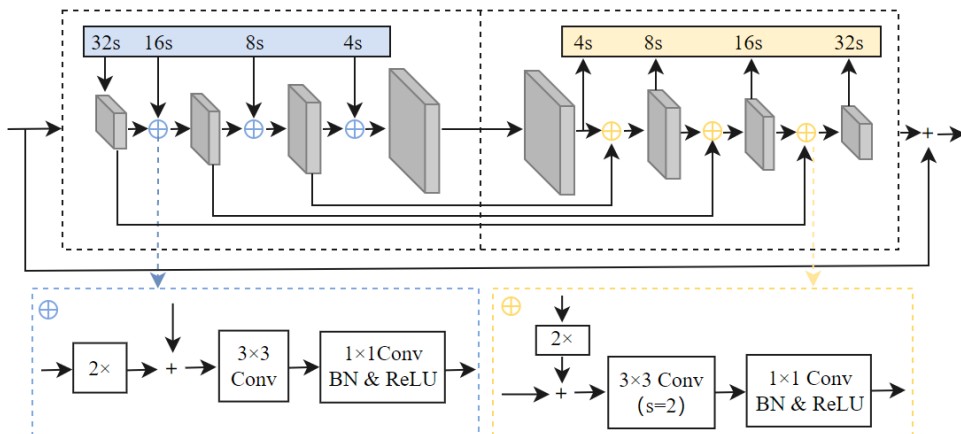

**Figure 2.** The feature refinement module named as FRM.

### 3.2. Positional Encoding for Transformer

Positional encoding is two-dimensional, which allows the attention mechanism to capture 2D spatial information more effectively. The output is then vectorized and input into the Encoder, as shown in Equations (2) and (3).

$$F' = F + P \tag{2}$$

$$E = Mat2Vec\left(F'\right) \tag{3}$$

where $P \in \mathbb{R}^{W \times H \times d}$ indicates positioning encoding, $F'$ is the feature adding location information, and $Mat2Vec(.)$ is a matrix-to-vector converter.

### 3.3. Horizontal Text Proposal Generation

The proposed CA-STD is based on Detection Transformer (DETR) [27], which is an end-to-end object detector predicting all objects at once and requiring less prior knowledge. The conditional attention mechanism, in which conditional spatial queries are introduced to improve localization and speed up the training process, is used to improve traditional Transformer.

The encoder aims to refine the content embedding from the backbone. The decoder consists of the following three main layers. A self-attention layer for removing repetitive predictions interacts with the output of the previous decoder layer. A cross-attention layer aggregates the encoder embedding to refine the decoder embedding. A feed-forward layer is used to predict the final categories and detection boxes.

In the original DETR, the cross-attention mechanism requires three inputs: query, key, and value. Key is formed by adding the content key $\mathbf{c}_k$ (the output of the encoder) and the positional key $\mathbf{p}_k$ (the position embedding of normalized two-dimensional coordinates). Same as $\mathbf{c}_k$, query is formed by content query $\mathbf{c}_q$ (the output of the self-attention layer) and spatial query $\mathbf{p}_q$ (i.e., object query $\mathbf{o}_q$). The attention weights are based on the dot product between query and key, as shown in in Equation (4):

$$
\begin{aligned}
&\left(\mathbf{c}_q + \mathbf{p}_q\right)^\top \left(\mathbf{c}_k + \mathbf{p}_k\right) \\
=&\mathbf{c}_q^\top \mathbf{c}_k + \mathbf{c}_q^\top \mathbf{p}_k + \mathbf{p}_q^\top \mathbf{c}_k + \mathbf{p}_q^\top \mathbf{p}_k \\
=&\mathbf{c}_q^\top \mathbf{c}_k + \mathbf{c}_q^\top \mathbf{p}_k + \mathbf{o}_q^\top \mathbf{c}_k + \mathbf{o}_q^\top \mathbf{p}_k
\end{aligned}
\tag{4}
$$

Conditional cross-attention module, shown as in Figure 3, decouples spacial and content information so that spacial query $\mathbf{p}_q$ and content query $\mathbf{c}_q$ focus on respective attention weights. In addition, it is notable that the $\mathbf{p}_q$ is computed based on the embedding

**f** from the previous decoder layer. Namely, the spatial information of different regions is determined by two factors, decoder embedding and reference point. After that, the information is mapped to the embedding space to form $\mathbf{p}_q$ so that the spatial queries are located in the same space mapped by the two-dimensional coordinates of the key. Therefore, the key is concatenated by $\mathbf{c}_k$ and $\mathbf{p}_k$, as shown in Equation (5).

$$\mathbf{c}_q^\top \mathbf{c}_k + \mathbf{p}_q^\top \mathbf{p}_k \tag{5}$$

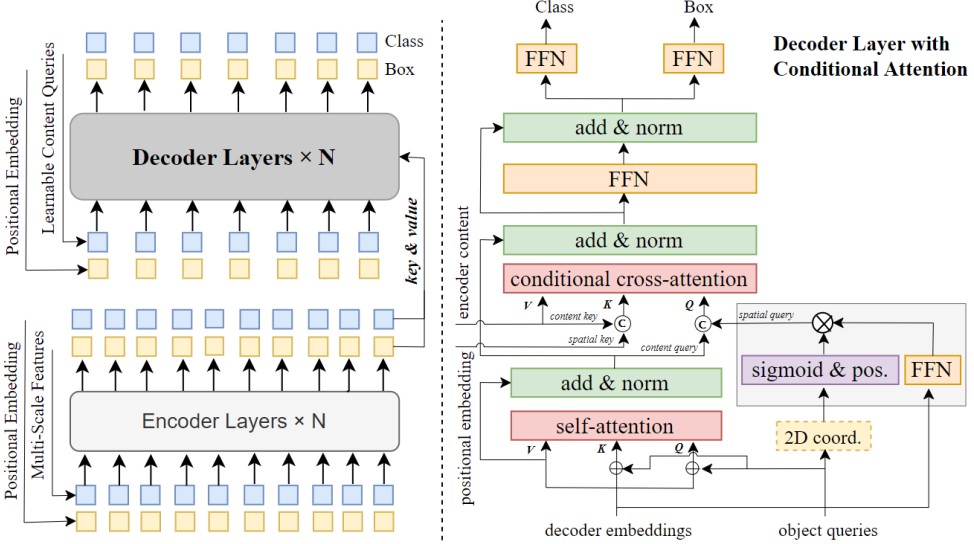

**Figure 3.** The Transformer with conditional attention.

The decoder's embedding contains the displacement of different regions relative to the reference point. The position prediction process consists of two steps: (1) predicts the box relative to the reference point in unnormalized space, and (2) normalizes the predicted box to the range [0, 1]. Among them, step (1) means that the decoder embedding **f** contains the displacement of the four points (forming the detection box) relative to the reference point **s** in the unnormalized space. Namely, both the embedding **f** and the reference point **s** are necessary to determine the spatial information of the different regions, the four points, and the region for predicting the classification score. The model predicts conditional spatial queries from the embedding **f** and the reference point **s**, as shown in Equation (6).

$$(\mathbf{s}, \mathbf{f}) \rightarrow \mathbf{p}_q \tag{6}$$

The reference points are obtained by regarding the unnormalized 2D coordinates as learnable parameters, and the unnormalized 2D coordinate is predicted from the object query $\mathbf{o}_q$. We normalize the reference point and then map it to a 256-dimensional sinusoidal positional embedding in the same way as the positional embedding for keys, as shown in Equation (7):

$$\mathbf{p}_s = \text{sinusoidal}(\text{sigmoid}(\mathbf{s})) \tag{7}$$

The conditional spatial query is computed by transforming the reference point in the embedding space. We choose the simple and computationally-efficient projection matrix, a diagonal matrix. The 256 diagonal elements are denoted as a vector $\lambda_q$. The conditional spatial query is computed by the element-wise multiplication, as shown in Equation (8):

$$\mathbf{p}_q = \mathbf{T}\mathbf{p}_s = \lambda_q \odot \mathbf{p}_s \tag{8}$$

### 3.4. Oriented Text Proposal Generation

In this module, each horizontal text is sampled uniformly $N_o$ along the contour line. Note that the horizontal text proposal represents the ground truth in the training phase and the prediction of boxes in the testing phase. After that, the new positions of these points are estimated by the Contour Localization Mechanism (CLM). Specifically, in the CLM, the contour feature extractor projects the contour point $X = \{x_i\}_{i=1}^{N_o}$ onto the feature $F_e$ to generate the semantic feature $F^{sem} \in R^{N_o \times D_e}$. Additionally, the location information of the contour point $F^{loc} \in R^{N_o \times 2}$ is formulated as $F_i^{loc} = x_i - x_{min}$, where $i$ denotes the index of the contour point and $x_{min}$ denotes the top-leftmost coordinate of the contour point. The semantic features $F^{sem}$ and the contour location information $F^{loc}$ will be concatenated to generate the original contour features $F_c \in R^{N_o \times (D_e+2)}$. Next, the Contour Information Aggregation (CIA) module takes $F_c$ as input to generate a more representative contour feature $F_{cia} \in R^{N_o \times D}$. The contour feature $F_{cia}$ is then fed into the Offset Prediction Head (OPH) to generate a contour point offset $O \in R^{N_o \times 2}$. Note that the OPH consists of three 1 × 1 convolutional layers (the first two layers are equipped with ReLU) with the number of filters being 256, 64, and 2, respectively. Next, new locations of contour points $X' \in R^{N_o \times 2}$ are obtained by $X + O$. Finally, the corner point generation module calculates the corner points of each text as $\mathbf{X}'[i * \lfloor N_o / N_c \rfloor]$, where $i \in \{0, 1, \ldots, N_c - 1\}$. $N_c$ is the number of corner points suggested by the oriented text. $\lfloor \cdot \rfloor$ denotes the floor operation. Therefore, the predicted points of all texts in each image can be referred to as $\widehat{\mathbf{X}}^{corner} \in \mathbb{R}^{N_t \times N_c \times 2}$. The loss function is expressed as Equation (9):

$$\mathcal{L}_{corner} = \frac{1}{N_t N_c} \sum_{i=1}^{N_t} \sum_{j=1}^{N_c} \mathcal{S}_{L_1}\left(\widehat{\mathbf{x}}_{ij}^{corner} - \mathbf{x}_{ij}^{corner}\right) \tag{9}$$

where $\mathbf{X}^{corner}$ denotes the label of the oriented detection boxes.

### 3.5. Arbitrary-Shape Text Contour Generation

In this stage, Contour Localization Mechanism (CLM) is used to gradually regress the oriented text proposal contours to obtain arbitrarily shaped text contours. Considering that the contours may evolve from some erroneous detections, there is a Reliable Contour Localization Mechanism (RCLM) to improve the confidence of the detected contours. The RCLM outputs the locations and confidence of the new contour point. Therefore, the loss function of the contour location evolution is expressed as Equation (10):

$$\mathcal{L}_{evolution} = \frac{1}{N_t N_a} \sum_{i=1}^{N_t} \sum_{j=1}^{N_a} \mathcal{S}_{L_1}\left(\widehat{\mathbf{x}}_{ij}^{final} - \mathbf{x}_{ij}^{final}\right) \tag{10}$$

where $\mathbf{X}_{i,j}^{final}$ is the $j$-th contour point of the $i$-th real text, which is sampled uniformly from the contour of an arbitrarily shaped scene text. Furthermore, the training objective of the contour scoring mechanism is considered as a text/non-text classification task, which is formulated as Equation (11):

$$\mathcal{L}_{csm} = -\frac{1}{N_t} \sum_{i=1}^{N_t} \log\left(\mathbf{s}_i^l\right) \tag{11}$$

where $l$ is the classification label of the contour line; $\mathbf{s}_i^l$ is the score of the region surrounded by the $i$-th contour line belonging to the background ($l = 0$) or the text ($l = 1$).

The contours of the scene text should form a closed shape. However, some points along the text contour are redundant, containing some noisy cues. Therefore, the module of the Contour Information Aggregation (CIA) is proposed to enrich the feature representation of contours, shown as Figure 4, where the original contour feature $F_c$ is first fed to a 9 × 9 circular convolution layer [30], followed by ReLU and batch normalization layer.

After that, seven CIA units with three different expansion rates are used to enhance the contour information with a multi-scale strategy because different expansion rates have different receptive field sizes. The first batch of normalization layers and the outputs of all CIA units are connected. The outputs of all CIA units are connected in series and passed through 1 × 1 D convolutional layer fused with 256 filters, followed by a max pooling operation. Finally, by concatenating the features of each contour point, the global pooled features are allocated to each contour point.

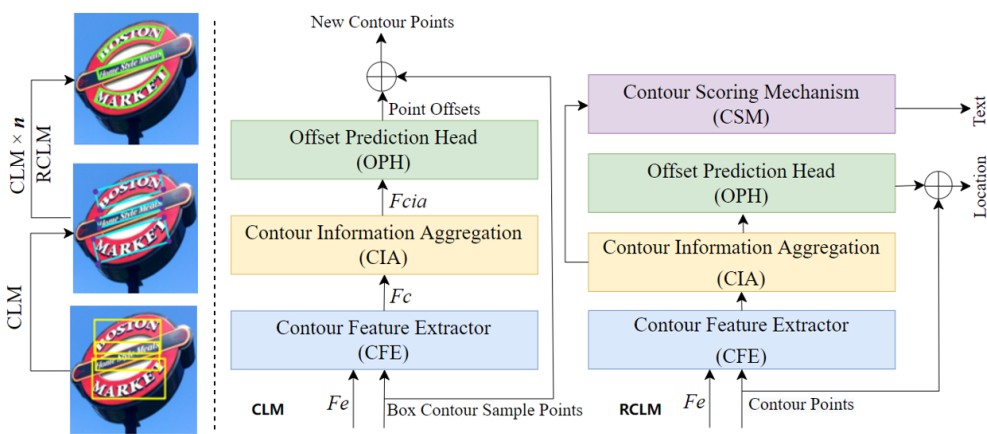

**Figure 4.** The process of fitting an arbitrarily shaped detection curve.

Reliable Contour Localization Mechanism (RCLM), as shown in Figure 4, is used to improve the reliability of the contour detected by the model. Specifically, the evolved contour points are sent to the contour feature extractor after CIA to generate the contour feature representation $F$ and then to the contour positioning branch to generate the final contour position $\widehat{\mathbf{X}}^{\text{final}}$. At the same time, the contour feature $\mathbf{F}_{cia}$ is also sent to RCLM to generate the contour score $s$, which is expressed as Equation (12):

$$\mathbf{s} = \varphi(\mathbf{F}_{cia}; \Theta_{csm}) \tag{12}$$

where $\varphi$ represents a contour scoring network, $\Theta_{csm}$ is the corresponding network parameter. Specifically, in $\varphi$, the input $\mathbf{F}_{cia}$ is first fed into a 1 × 1 convolution layer to obtain the feature representation $F_{csm}$. The average pooling operation $P_{avg}$ and the maximum pooling operation $P_{max}$ are used to generate the global feature representation $F'_{csm}$, which is expressed as $F'_{csm} = \left[P_{avg}(F_{csm}); P_{max}(F_{csm})\right]$. After that, three fully connected layers (hidden sizes of 512, 256, and 2) and one softmax are stacked to generate the final contour scores. Please note that the first two full connection layers are equipped with leaky-relu BN dropout operation, where the slope of leaky-relu is 0.2 and the dropout is 0.5.

To train a robust contour scoring network, it needs positive and negative samples to distinguish the contour of the scene text from the contour of the background. Specifically, the minimum bounding box of scene text with an arbitrary shape is considered as a positive sample. In addition, a negative sample mining technology is used to generate negative samples. To be specific, the outline of each scene text are placed on the image in the way of copying and moving. Next, the amount of overlap between the generated outline and all front outlines is calculated to assign the generated outline to different bins. Finally, the minimum bounding box of the contour randomly selected from the bin with the smallest degree of overlap is regarded as a negative sample.

*3.6. Loss Functions*

3.6.1. Matching Loss

$$\hat{\sigma} = \arg\min_{\sigma \in \mathfrak{S}_N} \sum_i^N \mathcal{L}_{match}\left(\hat{y}_{\sigma(i)}, y_i\right) \tag{13}$$

where $\mathcal{L}_{match}\left(\hat{y}_{\sigma(i)}, y_i\right)$ is a pair-wise matching cost between ground truth $y_i$ and a prediction with index $\sigma(i)$. This optimal assignment is computed efficiently with the Hungarian algorithm.

$$\mathcal{L}_{match}\left(\hat{y}_{\sigma(i)}, y_i\right) = -1_{\{c_i \neq \varnothing\}}\hat{p}_{\sigma(i)}(c_i) + 1_{\{c_i \neq \varnothing\}}\mathcal{L}_{box}\left(\hat{b}_{\sigma(i)}, b_i\right) \tag{14}$$

where $c_i$ indicates category label, $\hat{p}_{\sigma(i)}(c_i)$ indicates the probability that the prediction result is $c_i$ ; $b_i \in [0, 1]^4$ represents the label of the bounding box identified by the coordinates of the center point, width and height.

3.6.2. Hungarian Loss

$$\mathcal{L}_{\text{Hungarian}}(\hat{y}, y) = \sum_{i=1}^N \left[ -\log\hat{p}_{\hat{\sigma}(i)}[c_i] + 1_{\{c_i \neq \varnothing\}}\mathcal{L}_{box}\left[b_i, \hat{b}_{\hat{\sigma}}(i)\right] \right] \tag{15}$$

where $\hat{\sigma}$ is the optimal assignment computed in Equation (13). It is notable that the matching cost between an object and $\varnothing$ does not depend on the prediction, which means that in that case, the cost is a constant [31].

3.6.3. Bounding Box Loss

The L1 loss function, also known as the least absolute error, represents the absolute values of the difference between the target $b_i$ and the prediction $\hat{b}_{\sigma(i)}$.

$$\mathcal{L}_1\left(\hat{b}_{\sigma(i)}, b_i\right) = \| \hat{b}_{\sigma\|i\|} - b_i \|_1 \tag{16}$$

In the benchmarks of object detection, Intersection Over Union (IOU) is the most popular metric for evaluating models. However, optimizing the regression bounding box parameters' distance loss is not the same as maximizing the IOU. As a result, we utilize Generalized Intersection Over Union (GIOU) to calculate the regression loss of the bounding boxes, where IOU and GIOU are obtained from Equations (17) and (18):

$$IOU\left(\hat{b}_{\sigma(i)}, b_i\right) = \frac{Area\left(\hat{b}_{\sigma(i)} \cap b_i\right)}{Area\left(\hat{b}_{\sigma(i)} \cup b_i\right)} \tag{17}$$

$$GIOU\left(\hat{b}_{\sigma(i)}, b_i\right) = IOU\left(\hat{b}_{\sigma(i)}, b_i\right) - \frac{Area\left(C \smallsetminus \left(\hat{b}_{\sigma(i)} \cup b_i\right)\right)}{Area(C)} \tag{18}$$

where $Area(.)$ is the area of a set; $C$ denotes the smallest convex hull area that encloses both boxes $\hat{b}_{\sigma(i)}$ and $b_i$.

$$\mathcal{L}_{\text{GIOU}}\left(\hat{b}_{\sigma(i)}, b_i\right) = 1 - GIOU\left(\hat{b}_{\sigma(i)}, b_i\right) \tag{19}$$

$$\mathcal{L}_{box}\left(\hat{b}_{\sigma(i)}, b_i\right) = \lambda_1\mathcal{L}_1\left(\hat{b}_{\sigma(i)}, b_i\right) + \lambda_2\mathcal{L}_{\text{GIOU}}\left(\hat{b}_{\sigma(i)}, b_i\right) \tag{20}$$

where $\mathcal{L}_{box}\left(\hat{b}_{\sigma(i)}, b_i\right)$ represents the total loss caused by bounding boxes.

## 4. Experiments

Datasets, parameter setting, and evaluation metrics involved in the experiment will be detailed in the section. In addition, the effectiveness of the proposed model and its mod-

ules will be demonstrated through ablation experiments. Notably, TextFuseNet [12] and I3CL [25] were excluded from the state-of-the-art methods for the comparison experiments, even though they performed at optimal levels.

Regarding TextFuseNet [12], the model is based on character-level labels; in other words, training the model requires many character-level labels. However, it is well known that a large number of scene text datasets do not have character-level labels. In addition, labeling labels in character is time-consuming and expensive. Although a semi-supervised approach to generate character-level labels was proposed in the method, not only the strategy introduced a lot of computational costs, but also the model was trained in the direction of the results output by the semi-supervised model. However, the error between the labels output by the semi-supervised approach and the true labels of the text is a thorny problem in the current semi-supervised field. The proposed CA-STD does not require character-level labels to achieve the same detection results, saving a large amount of cost in annotation and computation.

In terms of I3CL [25], the authors proposed to treat texts as instances to be detected, and then modeled the relationship of different texts based on the gap features, so that instances belonging to the same text can be detected as a whole. The method mainly meets the challenge that the model detects larger instances belonging to the same text as a different instance. However, it can be seen that the model is based on the Convolutional Neural Network (CNN), which relies on traditional components such as Feature Pyramid Network (FPN) and Region Of Interest (ROI). The design of components requires a lot of a priori knowledge. In addition, the phased training of multiple components reduces the speed of the model, which makes it difficult to achieve real-time inference. The proposed CA-STD does not rely on CNN components and utilizes a Transformer in scene text detection, which is a major new trend.

### 4.1. Datasets

There are three datasets in the experiments. SynthText [32] is used to pre-train the model. TotalText [33] and CTW-1500 [34] are used to fine-tune and validate the model.

- **SynthText**. There are 858,750 synthetic pictures with character-level labels, word-level labels, and bounding box coordinates. A non-class is added to the original dataset to represent the boundary boxes without text.
- **TotalText**. The text is horizontal, random, and curved in the dataset. Additionally, 1255 images are used to train the model, and 300 images are used to test it. Only word-level labels and bounding boxes are included in the annotations file.
- **CTW-1500**. CTW-1500 is utilized to train a model for curved text, containing 1000 training images and 500 test images with word-level annotations and bounding boxes.

### 4.2. Implementation Details

This section is going to describe the experiments' implements in detail. There are the experimental environment, the setting of the model's hyper-parameters, the basis of the model evaluation, and the experimental results and analysis.

#### 4.2.1. Experimental Environment and Hyper-parameters

As described in the architecture, ResNet50 is regarded as the backbone and the Feature Refinement Module (FRM) is used to refine the feature map. There are six layers in both the encoder and decoder of the Transformer trained on NVIDIA GeForce RTX 3090 GPUs. The optimizer selected is AdamW [35] with a base learning rate of $2 \times 10^{-4}$, $\beta_1 = 0.9$, $\beta_2 = 0.999$, and weight decay of $10^{-4}$. The model is trained for 50 epochs, and the learning rate is decayed at the 40th epoch by a factor of 0.1.

#### 4.2.2. Evaluation Metrics

Precision (P), Recall (R), and the F1-score (F1) are the evaluation measures of the model. P denotes the model's precision, or the likelihood that the sample judged positive

by the model is just a positive example. The letter R denotes the model's capacity to find all positive classifications. F1-score as an evaluation indicator is used to balance Precision and Recall. The three measures listed above are commonly used to detect scene text.

### 4.2.3. Experiment Results

As can be seen from the overall pipeline of the model, the detection process is divided into two steps in total. In the first step, the improved Transformer is used to obtain rectangular detection boxes in the horizontal direction, which is equivalent to detecting the general area where the text is located. In the second step, the exact text position is obtained through information aggregation, and finally, the arbitrarily shaped detection boxes that fit the scene text are obtained.

There are two datasets, TotalText and CTW-1500, which are used to verify the model's effectiveness. The experimental results are shown in Tables 1 and 2, where the bold is used to indicate the maximum value of the column for ease of reading. The score of different models on TotalText is shown in Table 1. The precision of CA-STD reaches the peak 82.9, which is higher than for state-of-the-art methods. It proves that the proposed model can detect the scene text accurately. Table 2 shows the score of different models on CTW-1500. It can be seen that the proposed model can balance its recall while ensuring accuracy. Namely, the F1-score of the model achieves the maximum value on this dataset.

**Table 1.** The results on TotalText.

| Methods | P | R | F1 |
|---|---|---|---|
| SegLink [7] | 30.3 | 23.8 | 26.7 |
| DeconvNet [33] | 33.0 | 40.0 | 36.0 |
| MaskSpotter [8] | 55.0 | 69.0 | 61.3 |
| TextSnake [11] | 82.7 | 74.5 | 78.4 |
| PSENet [10] | 81.8 | 75.1 | 78.3 |
| PAN [36] | 79.4 | 88.0 | 83.5 |
| LOMO [37] | 75.7 | **88.6** | 81.6 |
| SAST [38] | 76.9 | 83.8 | 80.2 |
| CRNet [39] | 82.5 | 85.8 | **84.1** |
| CA-STD | **82.9** | 82.1 | 82.5 |

**Table 2.** The results on CTW-1500.

| Methods | P | R | F1 |
|---|---|---|---|
| SegLink [7] | 42.3 | 40.0 | 40.8 |
| CTD+TLOC [40] | 69.8 | 77.4 | 73.4 |
| TextSnake [11] | **85.3** | 67.9 | 75.6 |
| PSENet [10] | 80.6 | 75.6 | 78.0 |
| CRAFT [41] | 81.1 | 86.0 | 83.5 |
| LOMO [38] | 69.6 | **89.2** | 78.4 |
| SAST [13] | 77.1 | 85.3 | 81.0 |
| CA-STD | 83.1 | 84.5 | **83.8** |

### 4.3. Ablation Study

The ablation studies were conducted on Total-Text and CTW-1500 to verify the effectiveness of each proposed module in CA-STD. The experimental results are shown in Tables 3 and 4, where the bold is used to indicate the maximum value of the column for ease of reading. For each dataset, we trained four models by adding the proposed modules gradually. "Baseline" denotes the original model without any modules. "+FRM" denotes the model with the Feature Refinement Module (FRM). "+Cond-atte" denotes the model introducing the Conditional Attention Module but without FRM. "+Cond-atte(FRM)" denotes the model using a Transformer with Conditional Attention Module based on "+FRM". "+CLM/RCLM" denotes that Contour Localization Mechanism (CLM) and Reli-

able Contour Localization Mechanism (RCLM) are added to the model with "+Cond-atte". "+CLM/RCLM(FRM)" denotes that CLM and RCLM are added to the above model with FRM, namely, the proposed CA-STD.

As can be seen, "+FRM" improves the performance of the baseline model consistently on all two datasets, e.g., 0.9 and 0.4 gains in terms of the Recall on Total-Text and CTW-1500, respectively. In addition, integrating it with "+Cond-atte(FRM)" further brings absolute performance gains in terms of the F1-score increase by 1.0 and 0.9, respectively. By contrast, the combination of "+CML/RCLM(FRM)" achieves a better gain of 1.7 and 2.8 on F1-score.

From the experiments (first two rows of Tables 3 and 4), the model indeed performs worse on F1 than the Baseline methods after adding the FRM. However, the contribution made by the module on Recall should not be ignored, which is also critical to the model. Moreover, further ablation experiments demonstrate that the FRM improves the F1 score of the models based on modules "+Cond-atte" and "+CML/RCLM" by 0.4 and 0.5 on TotalText, and 0.7, 1.1 on CTW-1500, respectively. Therefore, the FRM is necessary and effective for the proposed CA-STD.

**Table 3.** The ablation study on TotalText.

| Methods | P | R | F1 |
|---|---|---|---|
| Baseline | 81.4 | 80.2 | 80.8 |
| +FRM | 80.3 | 81.1 | 80.7 |
| +Cond-atte | 81.5 | 81.4 | 81.4 |
| +Cond-atte(FRM) | 81.9 | 81.7 | 81.8 |
| +CLM/RCLM | 82.2 | 81.9 | 82.0 |
| +CLM/RCLM(FRM) | **82.9** | **82.1** | **82.5** |

**Table 4.** The ablation study on CTW-1500.

| Methods | P | R | F1 |
|---|---|---|---|
| Baseline | 82.3 | 79.8 | 81.0 |
| +FRM | 80.6 | 80.2 | 80.4 |
| +Cond-atte | 81.6 | 81.2 | 81.4 |
| +Cond-atte(FRM) | 82.5 | 81.4 | 81.9 |
| +CLM/RCLM | 82.8 | 82.6 | 82.7 |
| +CLM/RCLM(FRM) | **83.1** | **84.5** | **83.8** |

## 5. Discussion and Conclusions

Experiments show that the proposed method is capable of detecting scene text with an arbitrary shape, which is a huge leap in Scene Text Detect (STD). The detection box is the foundation for increasing text recognition performance in two-stage methods. However, there are still certain areas in which CA-STD may be improved because it is based on a two-stage strategy. Specifically, the stage of generating horizontal detection boxes and fitting curve is trained correspondingly, which has a certain impact on the performance and the speed of inference of the model. Therefore, the implementation of an end-to-end model based on the proposed model will be the focus of our subsequent research.

The proposed CA-STD, based on a conditional attention mechanism, can detect variably shaped scene text, which is critical for obtaining textual information from natural scenes. Existing STD methods fall short when dealing with the variation in text curvatures, orientations, and aspect ratios in complex backgrounds. Rather than a single rectangular detection box, CA-STD generates detection boxes consistent with the text. There is much noise for future text recognition under complex background conditions in that the rectangle detection area still contains a lot of background information. Therefore, being able to fit the shape of the text as closely as possible is very important for STD. In addition to the Feature Refinement Module (FRM), the Contour Information Aggregation (CIA) module is proposed to facilitate the evolution of the detection contours and obtain detection curves

with arbitrary shape. The quantitative and qualitative analysis proves that the CA-STD can detect variably shaped scene text effectively.

**Author Contributions:** Conceptualization, X.W. and Y.Q.; methodology, X.W.,Y.Q. and J.S.; software, X.W.; validation, X.W., Y.Q. and J.Y.; formal analysis, Y.Q., J.Y. and Y.W.; investigation, Y.Q. and Y.L.; resources, X.W. and Y.L.; data preprocessing, Y.Q., Y.H. and Q.Q.; writing—original draft preparation, X.W., Y.Q.; writing—review and editing, X.W. and Y.Q.; visualization, X.W. and Y.Q.; supervision, X.W., Y.H. and Q.Q. All authors have read and agreed to the published version of the manuscript.

**Funding:** This work was funded by the National Natural Science Foundation of China (Grant No. 62172267), the National Key R&D Program of China (Grant No. 2019YFE0190500), the Natural Science Foundation of Shanghai, China (Grant No. 20ZR1420400), the State Key Program of National Natural Science Foundation of China (Grant No. 61936001), the Shanghai Pujiang Program (Grant No. 21PJ1404200), the Key Research Project of Zhejiang Laboratory (No. 2021PE0AC02).

**Informed Consent Statement:** Informed consent was obtained from all subjects involved in the study.

**Data Availability Statement:** TotalText is publicly available at https://gitee.com/reatris/Total-Text-Dataset (accessed on 30 November 2022 ); CTW-1500 is publicly available at https://ctwdataset.github.io/ (accessed on 30 November 2022).

**Conflicts of Interest:** The authors declare no conflict of interest.

## Abbreviations

The following abbreviations are used in this manuscript:

| | |
|---|---|
| STD | Scene Text Detection |
| CA-STD | Conditional Attention-based Scene Text Detection |
| FRM | Feature Refinement Module |
| CIA | Contour Information Aggregation |
| CLM | Contour Localization Mechanism |
| RCLM | Reliable Contour Localization Mechanism |
| CNN | Convolutional Neural Network |
| ROI | Region Of Interest |
| RPN | Region Proposal Network |
| FPN | Feature Pyramid Network |
| MSER | Maximum Stable Extreme Regions |
| NMS | Maximum Suppression |
| IOU | Intersection Over Union |
| GIOU | Generalized Intersection Over Union |

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
