# Peer review of "CA-STD: Scene Text Detection in Arbitrary Shape Based on Conditional Attention"

_information, doi:10.3390/info13120565_

Round 1

Reviewer 1 Report

This paper conducted research on the topic of scene text classification using conditional attention. The main contribution is to generate the bounding boxes with arbitrary shapes by the Contour Information Aggregation. There are some errors that need to be checked:

1. line 28 should specify the equation before the  \ref of the equations.

2. there is no detail shown in Figure 2 for the condition attention, it should be pointed out.

3. there is a lack of ablation studies in the experiments, and the analysis of the result is not detailed.

Reviewer 2 Report

The authors propose a scene text detection framework to detect arbitrarily shaped text against complicated backgrounds. Specifically, they propose a Feature Refinement Module and a conditional attention mechanism, followed by a Contour Information Aggregation (CIA) step. They measure the performance of their model on two widely-adopted datasets.

The paper is certainly of high interest for recognizing arbitrarily-shaped texts in images. However, I have some major concerns:

  • Some ablation studies would be required to understand the role of the components you introduced, mainly the Feature Refinement Module (FRM) and the conditional attention.

  • You claim that you obtain state-of-the-art results. However, there are some works that outperform your numbers [1, 2] on both datasets. You should consider adding them and better explain in which ways your method differs.

Furthermore,

  • The addition of figures in the introduction and of qualitative examples in the Experiments section would help the overall understandability of the paper. Also, you should consider adding a high-level diagram containing the overall pipeline of your approach.

  • Figure 2 is somewhat hard to read. It would be better to find in the figure the same variables you defined in the equations to understand the various steps. Also, there are some input/output lines that have no name (like the arrows in the top-right, after the FFN blocks), so it is hard to understand what information they are carrying.

  • The paper is hard to read in some parts, due to some concepts that are unclear without some proper figures or examples (i.e., the reference point, or the contour points).

Minor comments:

  • Better specify what is “s” in Figure 1.

  • “as shown in 2 and 3” ->  “as shown in Equations 2 and 3”. There are other places in the text where this happens.

[1] Ye, Jian, et al. "TextFuseNet: Scene Text Detection with Richer Fused Features." IJCAI. Vol. 20. 2020.

[2] Du, Bo, et al. "I3CL: Intra-and Inter-Instance Collaborative Learning for Arbitrary-shaped Scene Text Detection." International Journal of Computer Vision (2022): 1-17.

Round 2

Reviewer 2 Report

The authors carefully addressed most of the comments, considerably improving the first version. However, in light of the newly added information, I still have a major concern:

  • From the added ablation study, it results that the FRM module worsens the Baseline method. This should be at least justified. It would be better if you add other two rows in Tables 3, 4 where you report +Cond-atte and +CLM/RCLM but without +FRM, commenting and justifying the results accordingly.

Minor comments:

  • The exclusion from experiments of the two SOTA methods that you added in the related works should be justified also in the manuscript (not only in the reviewer’s response letter).

  • As shown in 7 -> As shown in Equation 7
